# Automated kidney tumor segmentation with convolution and transformer network

Zhiqiang Shen[1], Hua Yang[2], Zhen Zhang[1], and Shaohua Zheng[1]

[1] College of Physics and Information Engineering, Fuzhou University, Fuzhou, China
[2] College of Photonic and Electronic Engineering, Fujian Normal University, Fuzhou, China
sunphen@fzu.edu,cn

**Abstract.** Kidney cancer is one of the most common malignancies worldwide. Early diagnosis is an effective way to reduce the mortality and automated segmentation of kidney tumor in computed tomography scans is an important way to assisted kidney cancer diagnosis. In this paper, we propose a convolution-and-transformer network (COTRNet) for end to end kidney, kidney tumor, and kidney cyst segmentation. COTRNet is an encoder-decoder architecture where the encoder and the decoder are connected by skip connections. The encoder consists of four convolution-transformer layers to learn multi-scale features which have local and global receptive fields crucial for accurate segmentation. In addition, we leverage pretrained weights and deep supervision to further improve segmentation performance. Experimental results on the 2021 kidney and kidney tumor segmentation challenge demonstrated that our method achieve dice scores of 92.28%, 55.28%, and 50.52% for kidney, masses, and tumor, respectively.

**Keywords:** Convolutional neural network · Kidney tumor · Transformer.

## 1 Introduction

Kidney cancer is one of the most common malignancies around the world leading to around 180000 deaths in 2020 [18]. Early diagnosis of kidney tumor is crucial to reduce kidney cancer mortality. Computed tomography (CT) is an effective tool for early detection and enable radiologists to study the relationship between tumor size, shape, and appearance and its prospects for treatment [11]. However, highly accurate kidney cancer diagnosis relies on the experience of doctors and the treatment subjective and imprecise. Computer-aid diagnosis (CAD) system can be used as a second observer to confirm the diagnosis and reduce the heavy burdens of radiologists.

Recently, deep learning-based CAD systems have been widely developed for cancer diagnosis and achieved great performance [22,15,25]. Yu et al. used the crossbar patches and the iteratively learning strategy to train two sub-models for kidney tumor segmentation [22]. Ozdemir et al. developed a CAD system for

pulmonary nodule segmentation and nodule-level and patient-level malignancy classification [15]. Zheng et al. designed a symmetrical dual-channel multi-scale encoder module in the encoding layer for colorectal tumor MRI image segmentation [25]. A CAD system of kidney cancer diagnosis may include kidney and kidney tumor segmentation as well as kidney cyst segmentation. This is a challenging task because the locations, textures, shapes, and sizes of kidney tumor are diverse in CT images as shown in Fig. 1. U-Net and its variants have been widely used for end to end lesion segmentation [17,3,26,9]. U-Net is an encoder-decoder architecture where the encoder and decoder are connected by the skip connections [17]. The encoder is with stacked local operators, i.e., convolutional layers and down-sampling operators, to aggregate long-range in-formation gradually by sacrificing spatial information. The decoder is with up-sampling and convolution layers to recover spatial resolution and refine the details. The skip connections transfer the features from the encoder to the corresponding layers of the decoder, which enable information reuse.

However, U-Net has limitations to explicitly model long-range dependency be-cause the convolution are local operators. To aggregate long-range information, the encoder usually stacks several convolutional layers interlaced with down-sampling operators. Long-range dependency, i.e., large receptive field, is crucial of a model to perform accurate segmentation. Therefore, previous researches improved the U-Net to overcome this limitation implicitly by stacking more convolution layers in the blocks of U-Net. For example, MultiResUNet designed a MultiResBlock with three convolution layers to learn multi-scale information [9]. However, large amount of convolution layers stacking in a model may influence its efficiency and cause the gradient vanish by impeding the back-propagation process.

In this paper, we propose a convolution-and-transformer network (COTR-Net) for end to end kidney, kidney tumor, and kidney cyst segmentation. COTR-Net has an encoder-decoder architecture where the encoder and the decoder are connected by the skip connections. To overcome the problem mentioned above, we inserted the transformer encoder layers [19] to the encoder of COTRNet. Specifically, the encoder consists of several convolutional layers interlaced with transformer encoder layers and max-pooling operators to explicitly model long-range dependency. The decoder is composed of several up-sampling operators each of which is followed by convolution layers to recover spatial resolution and refine contexture details. In addition, we leverage the pretrained ResNet [4] to develop the encoder, which accelerates the optimization process and prevent the model from falling into local optimum. Moreover, we added the deep supervision [12] to avoid the vanishing gradient phenomenon and rapidly train the model, in which the information between the final output and the side outputs is progressively aggregated. We evaluated the proposed method on the 2021 kidney and kidney tumor segmentation challenge (KITS21) [5]. Experimental results demonstrate the effectiveness of the proposed method.

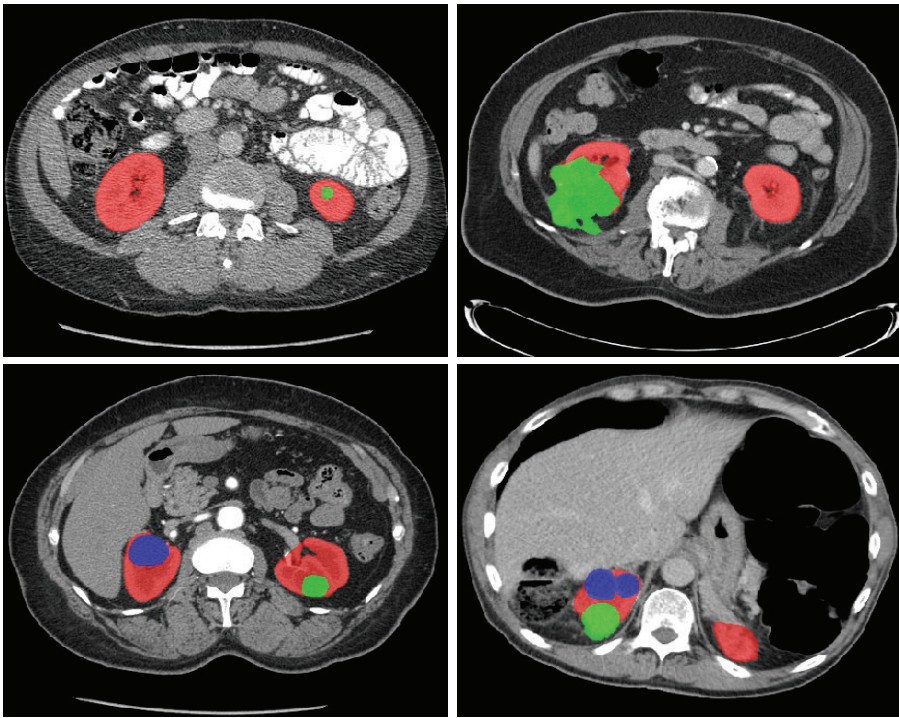

**Fig. 1.** Examples of an axial slice of kidney, tumor, and cyst with various locations, textures, shapes, and sizes in CT scans. kidney, tumor, and cyst are highlighted by red, green, and blue respectively.

## 2    Related work

In the following, we review the literature related to the proposed method on two aspects including deep learning-based medical image segmentation methods and self-attention mechanism.

### 2.1    Medical image segmentation

The emergence of U-Net has greatly promoted the development of medical image segmentation [17]. Then, 3D U-Net extended the vanilla U-Net to the 3D scenario. Since then, several new networks, such as U-Net++ [26], and MultiResUNet[9] both including 2D and 3D version of architectures has been proposed for medical image segmentation by improving U-Net architecture. In kidney tumor segmentation, although CT scans has 3D spatial attribute, this task can be resolved in 2D or 3D scenario. Jackson et al. proposed an automatic segmentation framework based 3D U-Net for kidney segmentation [10]. Hou et al. designed a triple-stage self-guided network to achieve accurate kidney tumor

segmentation [6]. Hu et al. presented a boundary-aware network with a shared 3D encoder, a 3D boundary decoder, and a 3D segmentation decoder for kidney and renal tumor segmentation [8].

Although processing using 3D data can reflect the whole information about the nodules, it will also require more training time and storage space. In addition, CT scans usually have different slice thicknesses, which are not recommended to be uniformly used in 3D segmentation task. On the contrary, 2D slices are not influenced by the slice thickness, and both training time and resources needed for processing are less than 3D patches. Therefore, in this work, we use 2D slice to perform the kidney tumor segmentation task.

### 2.2   Self-attention mechanism

Self-attention mechanism is an effective tool for convolution neural networks (CNN) to localize the most prominent area and capture global contextual information [19,20,7]. Oktay et al. proposed an attention U-Net where the attention gates are added to the skip connections to filter the features propagated through the skip connections [14]. Wang et al. designed a non-local U-Net for biomedical image segmentation, in which the non-local block was inserted into U-Net as size-preserving processes, as well as down-sampling and up-sampling layers [21]. Zheng et al. proposed a dual-attention V-network for pulmonary lobe segmentation where a novel dual-attention module to capture global contextual information and model the semantic dependencies in spatial and channel dimensions is introduced [24]. Recently, transformer has been exploited in medical image processing [23,2]. Zhang et al. presented a two-branch architecture, which combines transformers and CNNs in a parallel style for polyp segmentation [23]. Chen et al. proposed a TransUNet in which the transformer encodes tokenized image patches from a convolution neural network (CNN) feature map as the input sequence for extracting global contexts [2]. However, these methods need large-scale GPU memory and this will not feasible for common users. Hence, we proposed a lighted transformer-based segmentation framework which needs only 8G GPU memory for network training.

## 3   Methods

The diagram of the proposed method is illustrated in Fig. 2. We detail the network architecture on Section 3.1 and the loss function on Section 3.2. The preprocess and postprocess methods are presented on Section 3.3. Other implementation details are introduced in Section 3.4.

### 3.1   Network architecture

We proposed COTRNet for kidney and kidney tumor segmentation. The network architecture of COTRNet is shown in Fig. 2. COTRNet take slices of size $224 \times 224$ as input and output the segmentation mask having the same size as

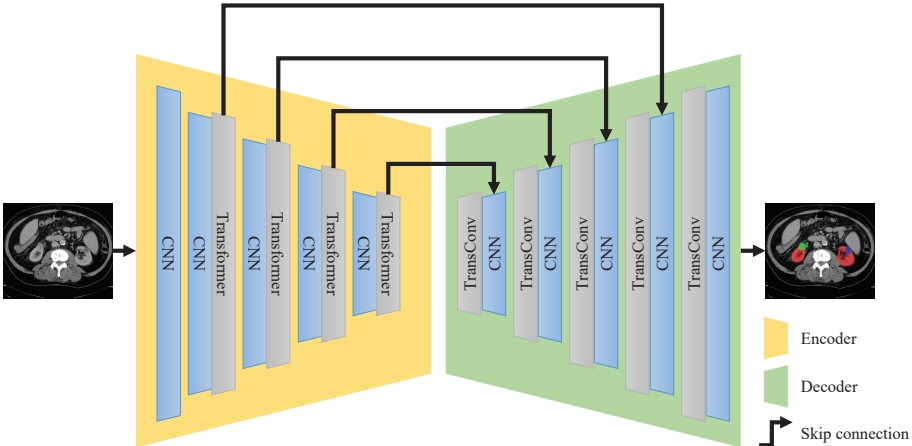

**Fig. 2.** The diagram of COTRNet.

input. The motivation of the proposed COTRNet is to capture long range dependencies i.e., large receptive field, of learned features for accurate kidney and tumor segmentation. Inspired by the detection transformer (DETR) which first exploited a pretrained CNN for feature extraction and transformer for feature encoding and predictions decoding [1]. The CNN and transformer are independent with each other in DETR. Although the transformer is proficient in learning global information, it takes the sequential data as input, which disentangles spatial structure of the input images.

Instead, we integrate the transformer with CNNs where the transformer layers are inserted into the CNNs to learning long range dependencies and the CNN then recover the spatial structure of the input images. COTRNet has an UNet-like architecture, which consists of an encoder, decoder, and the skip connections. Specifically, the encoder is composed of a series of convolution layers interleaved with transformer encoder layers. The transformer encoder layer is shown in Fig. 3. An input image is first transformed to low level features by the first convolution layer, then, to features has global information by the transformer encoder layer, and finally, the next convolution layer is utilized to reconstitute the spatial structure. Through a series of operations, the output features of the encoder are the fine-grained high-level representations which will then transfer to the decoder for refinement and segmentation predictions. Besides, feature maps transferred to the decoder through the skip connections also capture the global information of the input images, which further facilitate the decoder to predict segmentation masks via feature reuse.

To accelerate the training process and prevent the network from falling into local minimum, we leverage the pretrained parameters of ResNet18 to initialize the encoder of COTRNet. Moreover, we exploit deep supervision mechanism to promote grad-ually segmentation refinement, by supervising the hidden layers

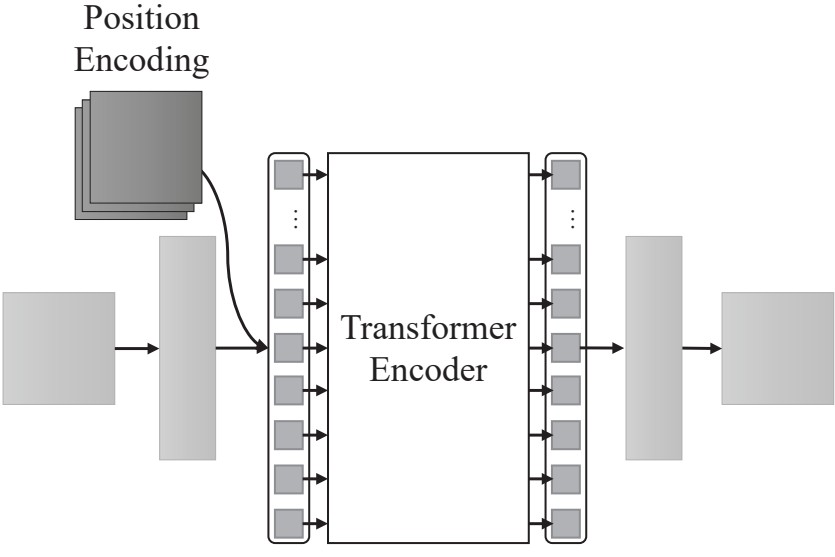

**Fig. 3.** The transformer encoder layer. A feature map is first transformed to a sequence, and then plus with position encoding to input into the transformer encoder layers.

to guide training by calculating the loss of the side outputs in the intermediate stages of the decoder.

### 3.2   Loss function

To overcome the data imbalance problem, we propose a class-aware weighted cross-entropy and dice (CA-WCEDCE) loss for kidney and kidney tumor segmentation. In general, the WCEDCE loss is a weighted combination of class weighted cross-entropy loss and class-weighted dice loss. The class-weighted cross-entropy (CWCE) loss is used to alleviate the inter-class unbalance problem, whereas the class-weighted dice (CWDCE) loss is exploited to solve the unbalance between each foreground class and the background class. Formally, the CA-WCEDCE loss is formulated as

$$
\begin{aligned}
\mathcal{L}_{CA-WCEDCE}(Y,\hat{Y}) &= \tfrac{1}{N}\sum_{i=1}^{N}\mathcal{L}_{CA-WCEDCE}\left(Y_i,\hat{Y}_i\right) = \\
&\tfrac{1}{N}\sum_{i=1}^{N}\left[\alpha\mathcal{L}_{CWCE}\left(Y_i,\hat{Y}_i\right) + (1-\alpha)\mathcal{L}_{CWDCE}\left(Y_i,\hat{Y}_i\right)\right]
\end{aligned}
\tag{1}
$$

where N is the batch size. $\alpha$ controls the contribution of the $\mathcal{L}_{CWCE}$ and $\mathcal{L}_{CWDCE}$ to the total loss $\mathcal{L}_{CAWCE}$. $Y_i$ is the $i_{th}$ ground truth of a batch of input images, and $\hat{Y}_i$ is the $i_{th}$ predicted mask of a batch of predictions.

The $\mathcal{L}_{CWCE}$ is represented as

$$\mathcal{L}_{CWCE}\left(Y_i,\hat{Y}_i\right) = \frac{1}{C}\sum_{c=1}^{C} w_c 1 - \sum_{j=1}^{M}[y_{j_c}\log\hat{y}_{j_c} + (1-y_{j_c})\log(1-\hat{y}_{j_c})] \quad (2)$$

And the $\mathcal{L}_{CWDCE}$ is denoted as

$$\mathcal{L}_{CWDCE}\left(Y_i,\hat{Y}_i\right) = \frac{1}{C}\sum_{c=1}^{C} w_c(1 - \frac{2\sum_{j=1}^{M} y_{jc}*\hat{y}_{jc}}{\sum_{j=1}^{M} y_{jc}+\hat{y}_{jc}}) \quad (3)$$

where $C$ refers the total number of classes which is equal to four (kidney, tumor, cyst, and the background) in our task. $M$ refers to the total number of pixels of the input slice in a batch. $w_c$ denotes the weighted coefficient of the $c_{th}$ class. $y_{j_c}$ is $j_{th}$ ground truth pixel of class $c$, and $\hat{y}_{j_c}$ is the corresponding predicted probability.

Since the deep supervision mechanism is exploited in network training, the total loss is formulated as

$$\mathcal{L} = \sum_{d=1}^{D} \beta_d \mathcal{L}_{CA-WCEDCE} \quad (4)$$

where $D$ is the total number of output decoder. $\beta_d$ is the weighted coefficient of the $d_{th}$ decoder.

### 3.3    Pre- and post- processing

Preprocessing. We perform data preprocessing follows four steps.

1. Normalization. The CT scans are clipped into [-200, 300] and normalized them into [0, 255].
2. Extraction. Slices that contain foreground regions of the normalized CT scans are extracted for network training according to the ground truth masks provided by KITS21 challenge.
3. Resample. The extracted slices are resized to [224, 224] according to the input size of the pretrained model.
4. Augmentation. Data augmentation including random flip, random rotation, random crop is utilized in training process.

Postprocessing. In inference, we conducted postprocess steps as follows.

1. Transformation. The logic predictions are transformed into probabilities through the softmax function. Then, we transform the probabilistic maps to the segmentation masks according to the maximum class.
2. Resample. We resize the segmentation masks to the original size according to the raw CT scans.
3. Integration. The whole predicted mask for a raw CT scan is obtained by combining all slice segmentation masks.
4. Refinement. Morphological operations are used to refined the segmentation masks.

### 3.4   Implementation details

We perform our experiments on PyTorch [16]. The models are trained via Adam optimizer with standard back-propagation with the learning rate of a fixed value of $1e-4$. We set the number of epochs as 20 and the batch size as 2. The networks are trained on a single NVIDIA GeForce GTX 1080 with 8G GPU memory.

In training, $\alpha$ is set as 0.5 to balance the contribution of CWCE loss and CWDCE loss. In Eq. 2, we set $w_1, w_2, w_3, w_4$, are set as 1, 2, 3, 4, respectively. In Eq. 4, $\beta_1 = 0.05, \beta_2 = 0.05, \beta_3 = 0.2, \beta_4 = 0.3, \beta_5 = 0.4$ to control the deep supervision mechanism. we random selected slices contained foreground objects as inputs. Data augmentation including random flip, random rotation, random crop was utilized in training process. In testing, all slices of a CT scan were input into the network to obtain predictions and the segmentation result of a CT scan was obtained by combining the predicted masks of all slices.

**Table 1.** Quantitative results on KITS21 training set through five-fold cross-validation. R:ResNet; T:Transformer encoder layer; P:Pretrained model; D:Deep supervision. SD: Surface Dice

| Method | Kidney (Dice) | Masses (Dice) | Tumor (Dice) | Kidney (SD) | Masses (SD) | Tumor (SD) |
|---|---|---|---|---|---|---|
| U-Net | 0.9015 | 0.4011 | 0.4148 | 0.8127 | 0.3017 | 0.3045 |
| U-Net+R+T | 0.9117 | 0.4655 | 0.4622 | 0.8598 | 0.3192 | 0.3093 |
| U-Net+R+P+D | 0.9169 | 0.4695 | 0.4838 | 0.8620 | 0.3331 | 0.3048 |
| U-Net+R+T+P | 0.9177 | 0.5082 | 0.5007 | 0.8742 | 0.3341 | 0.3435 |
| U-Net+R+T+P+D (COTRNet) | 0.9228 | 0.5528 | 0.5056 | 0.8853 | 0.3694 | 0.3548 |

## 4   Results

### 4.1   Dataset

We evaluated the proposed method on the KITS21 dataset. The KiTS21 dataset includes patients who underwent partial or radical nephrectomy for suspected renal malignancy between 2010 and 2020 at either an M Health Fairview or Cleveland Clinic medical center. KITS21 dataset includes 300 training cases of abdominal CT scans and the corresponding annotations of kidney, tumor, and cyst. The annotation files of each training case includes aggregated_AND_seg.nii.gz, aggregated_OR_seg.nii.gz, and aggregated_MAJ_seg.nii.gz. We eleveraged aggregated_MAJ_seg.nii.gz in our experiments. Note that we first perform the evaluation of our method on the training set because the test set is not publicly available.

### 4.2   Metrics

We used the same evaluation metrics as advocated by KiTS21 challenge, which include Sørensen-Dice and Surface Dice (SD) [13]. KITS21 leverages the hierarchical evaluation classes (HECs) to obtain a relative comprehensive measure. In

an HEC, classes that are considered subsets of another class are combined with that class for the purposes of computing a metric for the superset. The HEC of kidney and masses considers kidneys, tumors, and cyst as the foreground to compute segmentation performance; the HEC of kidney mass considers both tumor and cyst as the foreground classes; the HEC of tumor considers tumor as the foreground only.

### 4.3    Results on KITS21 training set

We reported the preliminary results on KITS21 challenge training set through five-fold cross-validation. All the methods are trained with the loss function presented in Section 3.2. Table 1 lists the quantitative results. In general, COTRNet outperforms other methods by a large margin, especially in tumor and cyst segmentation. Specifically, COTRNet achieved dice of 92.28%, 55.28%, and 50.56% for kidney, masses, and tumor, respectively; Measured by SD, COTRNet obtained 88.53%, 36.94%, and 35.48% for kidney, masses, and tumor, respectively. In these results, we can conclude that all components proposed in our methods contributed positively to the best performance. We also illustrate qualitative results on Fig. 4. As shown, COTRNet can accurately delineate the renal region, tumor, and cyst. Especially in cyst segmentation, although other methods overlooked the object regions, COTRNet can correctly locate the regions and delineate the margins of the cysts.

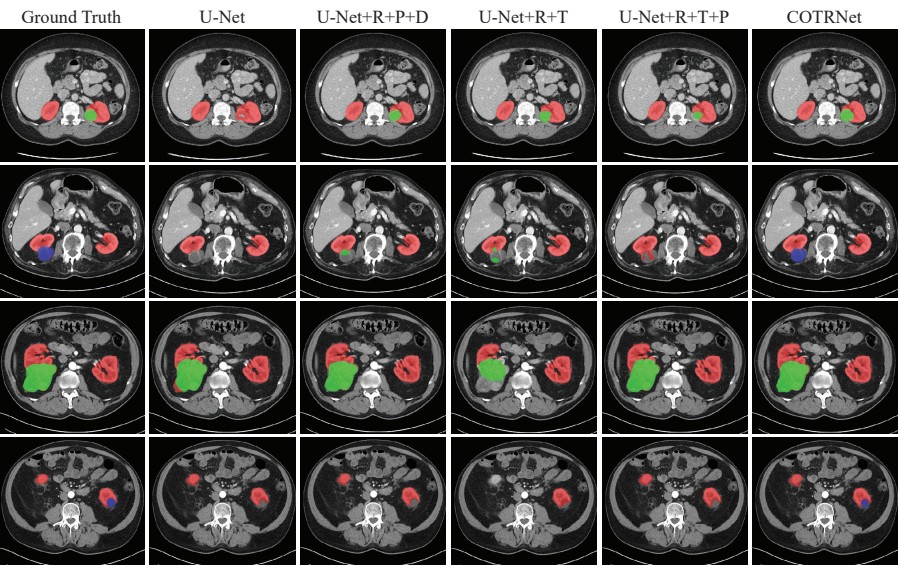

**Fig. 4.** Qualitative results on KITS21 dataset. R:ResNet; T:Transformer encoder layer; P:Pretrained model; D:Deep supervision.

## 5    Discussion and Conclusion

In this paper, we proposed the COTRNet to deal with kidney and tumor segmentation tasks. Inspired by the DETR that used transformer to model global information of features, COTRNet took advantage of transformer to capture long range dependencies for accurate tumor segmentation. Furthermore, we exploited pretrained parameters to accelerate convergence process. Deep supervision mechanism was used to gradually refine the segmentation results. We evaluated the proposed method on KITS21 dataset. COTRNet achieved comparable performance among kidney, cyst, and tumor segmentation. Experimental results demonstrated the effectiveness of the proposed method.

Although the transformer can explicitly model global information, it needs a large magnitude of GPU memory compared with the convolution operation. We will focus on reducing the memory consumption of transformer and developing more efficient and accurate segmentation framework in the future.

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
