# OpenReview forum: "Automated Kidney Tumor Segmentation with Convolution and Transformer Network"
_MICCAI.org/2021/Challenge/KiTS — Submitted to KiTS21 Challenge_

### Official Review · Reviewer_x4Y9 · 2021-08-30

**Rating:** 8

**Review:**

The authors present a transformer-based approach to the problem that they have termed COTRNet. The paper does a great job explaining it in detail, but one crucial detail is missing. How were the multiple annotations per instance aggregated in order to produce composite masks that were then used for training and validation? Most teams used majority voting (aggregated_MAJ), is this what you used as well? This should be made clear within the manuscript.

---

### Official Review · Reviewer_R4Rk · 2021-08-30

**Rating:** 9

**Review:**

This paper does an excellent job at providing an in-depth explanation of the approach used by this team for their submission. The authors also make effective use of figures and tables to improve clarity and support their arguments. I have no complaints or comments other than to request that the authors add their final results once they are known.

---

### Decision · Program_Chairs · 2021-08-30

**Decision:**

Minor Revisions

**Comment:**

Please address the reviewer comments and resubmit